# A Homologous Recombination System to Generate Epitope-Tagged Target Genes in *Chaetomium thermophilum*: A Genetic Approach to Investigate Native Thermostable Proteins

**DOI:** 10.3390/ijms23063198

**Published:** 2022-03-16

**Authors:** Nikola Kellner, Sabine Griesel, Ed Hurt

**Affiliations:** Biochemistry Center (BZH), University of Heidelberg, Im Neuenheimer Feld 328, 69120 Heidelberg, Germany; sabine.griesel@bzh.uni-heidelberg.de

**Keywords:** homologous recombination (HR), gene targeting, molecular genetic tools, thermophiliy, thermostable proteins, biodegradation

## Abstract

*Chaetomium thermophilum* is an attractive eukaryotic model organism which, due to its unusually high temperature tolerance (optimal growth at 50–52 °C), has a thermostable proteome that can be exploited for biochemical, structural and biotechnological applications. Site directed gene manipulation for the expression of labeled target genes is a desirable approach to study the structure and function of thermostable proteins and their organization in complexes, which has not been established for this thermophile yet. Here, we describe the development of a homologous recombination system to epitope-tag chromosomal genes of interest in *Chaetomium thermophilum* with the goal to exploit the derived thermostable fusion proteins for tandem-affinity purification. This genetic approach was facilitated by the engineering of suitable strains, in which factors of the non-homologous end-joining pathway were deleted, thereby improving the efficiency of homologous integration at specific gene loci. Following this strategy, we could demonstrate that gene tagging via homologous recombination improved the yield of purified bait proteins and co-precipitated factors, paving the way for related studies in fundamental research and industrial applications.

## 1. Introduction

Due to its lignocellulolytic lifestyle and high temperature tolerance, the fungus *Chaetomium thermophilum* exhibits a plethora of thermostable proteins, which includes enzymes applicable for various industrial utilizations. These comprise thermostable enzymes like proteases, chitinases, xylanases, cellulase, cellobiohydrolase or glucanases to be exploited for fast and efficient enzymatic degradation of lignocellulose and other biodegradation processes [1,2,3,4,5,6,7,8,9,10]. Accordingly, the thermophilic properties and hence increased stability of *C. thermophilum* proteins and protein complexes were exceedingly exploited for structural investigations, which led to detailed elucidations of fundamental cellular processes [11,12,13,14,15,16]. Today, a set of molecular tools has already been established in *C. thermophilum* in order to genetically manipulate this aspiring eukaryotic thermophile for various biochemical and biotechnological applications.

With the generation of thermostable selection markers, it was not only possible to establish a protocol for protoplast transformation of *C. thermophilum* [17], but also to develop a strategy to express fusion proteins with different affinity-tags, e.g., the ProtA- or Flag-tag [17], allowing their utilization in a wide range of biochemical techniques. This enabled for instance the expression of various fusion proteins for affinity purification of macromolecular protein complexes, in particular pre-ribosomal particles, and their subsequent high-resolution structural analysis by cryo electron microscopy [15,18,19]. Having several thermostable selection markers in hand, it was also possible to transform multiple plasmids and adapt the technique to a split-tag affinity purification system, which allowed to purify more complex and/or dynamic protein complexes under physiological conditions. In addition, we could identify novel interactions within distinct nuclear pore complex (NPC) modules, which revealed deeper insight into the NPC interaction network [20] and paved the way to unravel higher order structures during NPC assembly [13].

Disadvantageous to all of the aforementioned methods is, however, that only ectopic chromosomal integrations without any knowledge of the gene locus, number of integration events or expression levels were possible in the past. Although deletion mutants have already been created in *C. thermophilum* wild-type mycelia, the procedure appeared to be inefficient and imperfect (e.g., some desired strains were impossible to obtain despite screening thousands of candidate transformants) [21]. To circumvent these difficulties, we aimed to develop a homologous recombination (HR)-compatible *C. thermophilum* strain to conceive precise gene modification/replacement experiments, as HR allows efficient incorporation, expression control, and modification of desired target genes, respectively. To render *C. thermophilum* amenable for such genetic manipulation, we sought to inactivate the non-homologous random recombination activity, a common strategy to increase HR efficiency in related fungi [22,23,24]. During non-homologous end-joining (NHEJ), DNA double strand breaks are repaired without the need for any homology mediated by a protein complex consisting of the DNA-binding Ku70-Ku80 heterodimer, the DNA-dependent protein kinase DNA-PKcs, the exonuclease Artemis and the DNA ligase IV complex [25]. Hence, inactivation of either the *ku* genes or the *lig4* gene should lead to an increase in the rate of HR events. A respective approach has been successfully applied in closely related species, i.e., in *Chaetomium globosum* by deletion of the corresponding *ligD* gene, which resulted in a 70% success rate for targeted HR events versus 5% in the wild-type [26] and in *Aspergillus oryzae* by deletion of the LigD equivalent, which even achieved HR-rates of 100% in comparison to less than 30% in the wild-type [27].

The availability and refinement of the complete *C. thermophilum* genome sequence [28,29,30] enabled us to identify the corresponding components of the NHEJ pathway in the thermophile. Based on these findings, we could develop a strategy for in locus gene targeting in the thermophilic fungus *C. thermophilum*, allowing controlled expression of target genes, suitable for subsequent biochemical characterization of thermostable pre-ribosomal particles and nuclear pore complexes. As higher yields of purified protein complexes could be obtained in comparison to purifications from strains carrying ectopically integrated fusion constructs, our method promises an improved applicability for future investigations.

## 2. Results

### 2.1. Engineering HR-Compatibility in C. thermophilum

Initially we aimed to identify the thermophilic homologue of the *C. globosum* LigD protein, as its deletion should increase DNA repair by homologous recombination (HR). A BLASTP search (*C. thermophilum* genome resource, http://ct.bork.embl.de/, last accessed 1 February 2022) revealed the predicted *C. thermophilum* protein CTHT_0026720, which is conserved among eukaryotes and exhibits 69% identity and 79% similarity with the *C. globosum* LigD protein CHGG_03097 (UniProtKG accession number Q2H9K7). To obtain a *ligD* deletion mutant in *C. thermophilum*, a PCR-based approach according to Kämper [31] was used and the open reading frame was replaced by a hygromycin resistance cassette [15]. A linear construct consisting of a thermostable hygromycin phosphotransferase expression construct surrounded by 1.5 kb fragments homologous to the flanking regions of the *ligD* gene (Figure 1a) was used for the transformation of wild-type *C. thermophilum* protoplasts. 125 hygromycin resistant transformants were screened and subjected to PCR analysis with a combination of oligonucleotides specific for in locus integration of the transformed construct. About 40% of the obtained clones displayed an amplicon of the expected size, indicative of a correct gene deletion event; most mycelia, however, appeared to be heterokaryotic as the wild-type *ligD* allele could also be amplified with corresponding oligonucleotides. Only the PCR fragments of one clone (Δ*ligD#56*) implied a correct recombination event and the absence of an extra intact copy of the *ligD* open reading frame in the mycelia (Figure 1b). Accordingly, Δ*ligD#56* was chosen for further characterization. In order to confirm the correct deletion of *ligD*, Southern analysis was conducted. Therefore, ascospores were generated from strain Δ*ligD#56* (from now on called Δ*ligD)* and mycelia from four individual ascospores were harvested and used for the isolation of gDNA to be treated with restriction endonucleases for Southern blotting (Figure 1c). All strains showed the expected *Eco*RV fragments indicative of *ligD* deletion. The deletion of *ligD* had no apparent influence on mycelial growth during standard growth conditions. Growth on solid media supplemented with DNA double-strand break inducing agents, putatively to be affected in mutant strains deficient in NHEJ-activity, showed no obvious phenotype on methyl methanesulfonate (MMS) and a very mild growth phenotype on phleomycin (Appendix A).

As an alternative approach, we constructed a *ku70* disruption strain using the analogous methodology. In silico analysis revealed that the thermophilic ku70 protein is encoded by the gene CTHT_0032510. Accordingly, the *ku70* deletion mutant was constructed in the same way by transforming a linear DNA fragment containing the 1.5 kb sequences 5’and 3´ of the *ku70* open reading frame (CTHT_0032510), that are flanking a hygromycin resistance cassette. We could obtain several transformants showing the correct replacement of the *ku70* ORF within the chromosomal target DNA. Following ascospore formation and selection on hygromycin, individual mycelia were analyzed by PCR, corroborating the disruption of the *ku70* gene (Figure 1d). However, the resulting strains exhibited a mild growth defect on our standard cultivation media (Appendix A).

The resulting Δ*ligD* and Δ*ku70* strains were used for subsequent experiments and analyzed in terms of transformation rates and homologous recombination efficiency.

### 2.2. Utilization of the ΔligD Strain for in Locus Gene Targeting

In order to construct epitope-tagged target genes in *C. thermophilum* for affinity purification and subsequent analysis of thermostable protein complexes, we created C-terminal Flag-TEV protease cleavage site-Prot**A** (FpA) gene fusion constructs for homologous recombination in the *ligD* disruption strain. As proof-of-principle, we selected C-terminal FpA-fusions to the 90S pre-ribosome biogenesis factor Utp6 and the nuclear pore complex protein Nup82, respectively, as these two proteins have been studied in *C. thermophilum* previously [17,18]. To this end, an FpA-affinity tag in combination with a terbinafine resistance cassette (FpA-ERG1) was fused to a 1.0–1.5 kb fragment comprising the C-terminus of the respective open reading frame and the 1.5 kb homologous region downstream of the stop codon. Plasmid-based (*nup82*) or PCR-generated linear DNA (*utp6*) was used for transformation of *C. thermophilum* wild-type and Δ*ligD* protoplasts (see Materials and Methods for details). These constructs were finally used to assess the efficiency of homologous recombination (HR) in the *ligD* and *ku70* disruption strains and compare it to HR rates of the wild-type.

Homologous recombination events were initially confirmed by Western analysis since the expression of the respective FpA-tagged fusion protein is expected to be detectable only upon correct integration of the constructs into the corresponding gene locus (Figure 2a). We observed that for Utp6-FpA fusions, the transformation efficiency in Δ*ligD* was close to the one of wild-type protoplasts. However, the HR efficiency, i.e., integration at the *utp6* locus, was 0% in the wild-type (0/20 transformants were tested positive), whereas in the Δ*ligD* strain approximately 10% of the transformants (3/31) were tested positive by Western analysis. For Utp6-FpA fusions we frequently observed bands of a lower molecular weight that do not correspond to the size of full-length Utp6-FpA proteins, but possibly arise due to a possible internal AUG in the homologous region comprising the C-terminus of the Utp6 open reading frame, putatively being used as an alternative start codon. In the case of the Nup82-FpA fusion construct the transformation efficiency in Δ*ligD* was significantly lower than the one of wild-type protoplasts, as expected for NHEJ-deficient strains of related fungi [22]. In contrast, the percentage of successful HR events was 50% in Δ*ligD* (2/4 transformants were tested positive by Western blot) as compared to approximately 7% in the wild-type (3/38).

The correct homologous integration of the constructs was further verified by PCR- and Southern analysis (Figure 2). For PCR analysis, a combination of two oligonucleotides specific for in locus integration of the gene fusion constructs was applied that resulted in an amplicon of a designated size only in case of correct homologous recombination events. Southern analysis was additionally performed in order to differentiate between homo- or heterokaryotic mycelia, which cannot be accomplished by PCR analysis. For in locus Utp6-FpA gene fusion events it was observed that all positive strains were heterokaryotic, as they contained wild-type as well as Utp6-FpA fusion alleles, a common situation in multinucleate fungal mycelia (Figure 2c (“org”)). To circumvent this problem, we sought to produce sexual ascospores, a developmental process during which a mononucleate and haploid stage is passed, thereby segregating the different alleles and creating isogenic strains. As *C. thermophilum* belongs to the group of fungi that undergoes a homothallic life cycle, the formation of sexual ascospores is possible via selfing without the need for a matching mating partner. Upon germination of the ascospores of strain Δ*ligD utp6-FpA#25*, individual terbinafine-resistant (Terb^R^) and terbinafine-sensitive (Terb^−^) mycelia were isolated and again subjected to Southern analysis. We observed that 50% of the Terb^R^ mycelia now displayed DNA fragments corresponding to an isogenic Utp6-tagged strain, whereas the other half was still heterokaryotic. Terb^−^ mycelia exclusively exhibited the wild-type *utp6* allele (Figure 2c). Similar observations were made for the generation of the Δ*ligD Nup82-FpA* fusion strains. Southern analysis of the two original strains revealed one strain (*Nup82-FpA#1)* being heterokaryotic with the wild-type *nup82* allele still present, and the other strain (*Nup82-FpA#3)* putatively being homokaryotic without the corresponding wild-type fragment present (Figure 2d). Mycelia derived from Terb^R^ ascospores were again isolated and analyzed by PCR and Southern analysis and showed only fragments corresponding to the Nup82-tagged allele, whereas wild-type alleles were not detectable, indicative for the successful construction of isogenic strains (Figure 2d). Finally, correct in locus integration and replacement of the wild-type *nup82* gene with the FpA-tagged *nup82* variant in the Δ*ligD* strain could be further confirmed by Western analysis. In order to test the expression of the respective fusion protein, we used an α-*ct*Nup82 antibody that was previously shown to generate a specific nuclear rim staining in immunofluorescence analysis of *C. thermophilum* hyphae [17]. Mycelia from six individual Terb^R^ ascospores of each original positive strain were harvested and subjected to Western blotting, revealing bands of approximately 150 kDa, corresponding to the designated Nup82-FpA fusion proteins, whereas no wild-type Nup82 protein was detectable in the Δ*ligD* Nup82-FpA strains (Figure 2e).

Both the Utp6-FpA and the Nup82-FpA fusion constructs were also transformed into the *ku70* disruption strain in the same manner to assess the transformation efficiency and HR-frequency in this genetic background. For both constructs the transformation efficiency was significantly lower than compared to wild-type protoplasts, as expected for a NHEJ-deficient strain. Since all resistant mycelia showed correct integration of the fusion constructs in the respective genomic locus, indicative of high rates of homologous recombination, the Δ*ku70* strain may also be used for subsequent gene replacement studies in *C. thermophilum*, although a mild growth defect has to be taken into account.

In conclusion, we demonstrated the deletion of a LigD-equivalent ligase and the Ku70 protein, respectively, required for NHEJ-activity in *C. thermophilum* and established in locus gene fusion experiments driven by improved rates of homologous recombination. Even though we observed diverging results in terms of transformation efficiency and HR-rates, varying between different constructs, we could demonstrate that isogenic strains can finally be selected upon sexual ascospore formation in order to segregate the different alleles frequently observed in the heterokaryotic mycelia from *C. thermophilum*.

### 2.3. Biochemical Analysis of HR-Derived Strains Obtained by in Locus Gene Targeting

The respective affinity-purified protein complexes were analyzed and we compared the HR-derived in locus C-terminally tagged FpA-fusion proteins with the corresponding constructs that were ectopically integrated into the *C. thermophilum* genome. To this end, we purified Utp6-FpA-tagged proteins from Δ*ligD Utp6-FpA#25.10*, an isogenic HR-strain isolated from an individual ascospore, and CT13 [18] harboring random integrations of a plasmid-based Utp6-FpA-construct. From both strains, a typical early 90S pre-ribosomal particle pattern, which reflects the associated ribosome biogenesis factors, could be discerned when the final eluates were analyzed by SDS-PAGE and Coomassie staining (Figure 3a). A growth assay to evaluate the mycelial growth of the different Utp6-FpA strains was conducted but no obvious growth phenotype was detected (Appendix A). To examine Nup82-FpA fusion proteins, we affinity-purified the two positive HR-derived strains, #1 being heterokaryotic and #3 being an isogenic homokaryotic strain, respectively, and compared them to CT7 with random integrations of the plasmid-based Nup82-FpA-construct [17]. From all three strains the expected Nup82-Nup159-Nsp1 nuclear pore subcomplex was purified together with under-stoichiometrically associated members of the Nup84-subcomplex (Figure 3b) that has previously been observed and studied in *C. thermophilum* [13,17].

In both examples, the yield of the in locus FpA-tagged strain was reproducibly increased when equal amounts of affinity-purified protein lysates were compared from the respective HR-derived strains with the comparative strains harboring ectopically integrated fusion constructs. This might be due to the fact that inside the HR-derived strains the tagged proteins are not competing with the wild-type untagged protein for incorporation into their complexes (see Discussion). Taken together, our approach promotes precise genetic manipulation of *C. thermophilum* and points toward superior features of HR-based procedures over the conventional methods for gene tagging used until now.

## 3. Discussion

In this study, we established a method for producing thermostable fusion proteins expressed from their genomic loci and under their authentic promoters, leading to increased yields for tandem-affinity purification of eukaryotic macromolecular assemblies such as pre-ribosomal particles or nuclear pore subcomplexes. The preceding deletion of factors of the non-homologous end-joining (NHEJ) machinery in *Chaetomium thermophilum* resulted in tester strains exhibiting improved rates of homologous recombination, which facilitated the generation of subsequent strains carrying epitope-tagged gene fusion constructs at their authentic chromosomal loci.

In filamentous fungi transformed DNA is mostly integrated ectopically into the chromosomal DNA, mainly driven by NHEJ, and site-specific recombination occurs only at low frequencies. Targeting genes at their endogenous loci, however, allows to study specific gene functions, either by deletion of a target gene of interest, which is substituted by a marker gene or resistance cassette, or by integration of foreign DNA sequences inserted into the target gene, like affinity tags or other reporter constructs. To date, *C. thermophilum* was not amenable to the aforementioned methods and transformed DNA was randomly integrated into the genome. Because the targeted genomic loci remained unknown, functionally important genes might be disrupted or affected, possibly leading to growth defects and although gene expression of integrative constructs may be driven by the endogenous promoter of a given gene, the number of integrations cannot be controlled, posing a risk for overexpression artefacts. There are many examples of related organisms, where the use of mutant strains lacking components of NHEJ, namely deletion mutants of *ku70/ku80*, encoding DNA binding proteins required for NHEJ, or of *ligD*, encoding a DNA ligase essential for joining DNA double strand breaks during NHEJ, significantly increased the frequency of homologous recombination, e.g., in *Chaetomium globosum* [26], *Aspergillus* species [22,27,32,33], *Neurospora crassa* [23,34] or *Magnaporthe grisea* [35,36], thereby facilitating the controlled expression of target genes.

Upon replacing either *ligD* or *ku70* with a hygromycin resistance cassette in *C. thermophilum*, we analyzed homologous recombination rates to assess the efficiency of gene targeting at endogenous loci. Similar to other filamentous fungi, gene targeting is inefficient in wild-type *C. thermophilum*, which however could be improved in strains with disrupted NHEJ components. Whereas the *ku70* deletion strain showed reduced growth on standard cultivation media, possibly mirrored by the decreased DNA repair capability, no phenotypic defects in vegetative growth were observed for the *ligD* deletion strain, making it a decent recipient strain for transformation and valuable tool for gene targeting in *C. thermophilum*. Depending on the constructs used, recombination rates were lower as compared to related ascomycetes and varied between 10 and 50%. It was shown for a number of relatives that the length of homologous flanking sequence is fundamental for efficient site-specific recombination and generally, sequences of 1000 bp or more are sufficient for effective gene targeting in NHEJ-deficient recipient strains [22,23,24]. To further improve the efficiency of homologous recombination in *C. thermophilum*, the optimal length of the flanking sequence has to be determined systematically and analyzed more carefully. As there are examples of marker genes flanked by up to 4 kb of homologous regions [37], it will be worthwhile to gradually increase the length of flanking sequences and evaluate gene targeting frequencies in a substantially high number of transformants.

The multinucleate nature of the filamentous *C. thermophilum* mycelium is a considerable problem for genetic analyses and the presence of multiple nuclei in the protoplasts impedes the isolation of homokaryotic transformants. However, we could demonstrate that upon the generation of sexual ascospores and selection of individual spores, it is possible to segregate wild-type and recombinant alleles and hence to isolate isogenic strains harboring gene replacements or insertions.

The availability of the complete *C. thermophilum* genome sequence in combination with our gene targeting system now enables reverse genetics and allows functional genomic approaches. It was shown in a similar way involving the deletion of a *ligD*-homologue in the mesophilic relative *C. globosum* that targeted disruptions of epigenetic transcriptional regulators could be achieved, leading to the characterization of yet unexplored secondary metabolite gene clusters [26]. Analogous approaches might be useful for several biotechnological aspects of *C. thermophilum*. Not only are systematic gene deletions, which are important to unravel novel functions of genes, possible, but also the expression of truncated gene derivatives in case the gene is essential and a complete deletion is deleterious for the fungus, or the analysis of mutants in the endogenous gene locus. Additionally, new auxotrophic resistance markers may be conceived for *C. thermophilum*, which is of particular importance as many heterologous dominant resistance markers derived from mesophilic organisms do not withstand the high temperatures needed for cultivation.

Affinity-tagged fusion proteins expressed at the endogenous loci allow biochemical and subsequent structural investigation of thermophilic protein complexes under physiological conditions, thereby evading overexpression and other undesired consequences caused by random integration events. We could further demonstrate that the yield of protein purified from isogenic in locus tagged strains was consistently higher than the yield of protein purified from strains with ectopically integrated constructs, where untagged wild-type alleles of the genes are typically present, indicative of superior characteristics of our method for biochemistry. We explain the lower yield of the ectopically integrated construct because the presence of the cognate wild-type protein also integrates in the same subcomplexes and hence is competitive.

Other potential adaptations with in locus gene tagging being beneficial are fusions with a thermostable YFP variant (FFTS-YFP) for life cell imaging and subcellular localization studies [38] or fusion with a BioID reporter to screen for novel protein interactions [39]. Both markers have been proven to be functional under thermophilic conditions in *C. thermophilum* (unpublished results). Promoter fusions with different reporter genes to explore gene regulation/differential gene expression in *C. thermophilum* are likewise conceivable.

In conclusion, homologous recombination is a versatile method to manipulate gene expression in filamentous fungi and our advance to develop a gene targeting system with improved rates of homologous recombination now allows the controlled expression of target genes in *C. thermophilum*. As this thermophile evolved into a renowned model organism for biochemical and structural biology in recent years, our methods may open up new opportunities for genetic analyses and the characterization of thermostable proteins and complexes under native conditions. Due to its exceptional properties, *C. thermophilum* has also caused a sensation in industry with various biotechnological applications, e.g., as producer of thermostable enzymes for degradation of lignocellulosic plant material for the production of second-generation biofuels and other processes requiring enzymatic activity at high temperatures. For industrial applications the controlled genetic manipulation of organisms is not only of particular importance to meet any GMO safety standards, but also facilitates the controlled production of enzymes and other metabolites.

## 4. Materials and Methods

### 4.1. Strains and Growth Conditions

Wild-type *Chaetomium thermophilum var. thermophilum* (La Touche) [40] DSMZ 1495 was used. Growth conditions and media for the cultivation of *C. thermophilum* as well as the transformation of *C. thermophilum* protoplasts followed the protocols described previously [15,17].

Concisely, for transformation, protoplasts were generated from a submerged culture upon digestion of fungal mycelium with lysing enzymes from *Trichoderma harzianum* (Sigma-Aldrich, Taufkirchen, Germany, cat. no. L1412) for 3–4 h. The resulting protoplasts were filtered, washed and subsequently transformed with 5–10 µg of DNA. The transformed protoplasts were then directly plated onto solid regeneration media containing either 200 mg/mL hygromycin B (Sigma-Aldrich cat. no. H0654) or 1 mg/mL terbinafine hydrochloride (Sigma Aldrich cat. no. T8826), respectively. Hygromycin containing plates were cultivated at 42 °C to ensure the functionality of the resistance marker. The order of steps that have been taken to generate in locus tagged strains in *C. thermophilum* is visualized in Figure 4.

### 4.2. DNA Procedures

Molecular cloning techniques were based on protocols described by Sambrook and colleagues [41]. *E. coli* DH5α was used for plasmid propagation via standard procedures. Genomic DNA from *C. thermophilum* was isolated via phenol-chloroform extraction according to Al-Samarrai and Schmid [42].

For the deletion of *ligD* (CTHT_0026720) and *ku70* (CTHT_0032510), a PCR-based approach was adapted from Kämper [31]. Open reading frames were replaced by a hygromycin resistance cassette that was ligated to PCR-amplified 1.5 kb flanking sequences needed for homologous recombination. Using nested oligonucleotides to confer specificity, constructs were PCR-amplified and used for transformation of *C. thermophilum*.

C-terminal gene fusion constructs were generated by the ligation of the particular flanking sequences to an insertion cassette comprising the Flag-ProtA followed by a 300 bp sequence downstream of the gene encoding GAPDH (CTHT_0004880) to terminate transcription as well as a terbinafine resistance cassette [17]. Fusion constructs were subcloned for sequencing and either PCR-amplified linear DNA or linearized plasmid DNA was used for transformation. All *C. thermophilum* strains generated in this study are listed in Table 1.

Homologous integration of the constructs was verified by Southern analysis [43]. Therefore, the DIG High Prime DNA Labeling and Detection Starter Kit II (Roche, Mannheim, Germany, cat. no. 11585614910) was used according to the manufacturer’s instructions. Digoxigenin-labelled PCR-amplified flanking sequences were used as probes. All sequences of oligonucleotides used for PCR are listed in Table 2.

### 4.3. Expression and Affinity Purification of Proteins from C. thermophilum

Expression of affinity-tagged fusion proteins was confirmed with immunoblots of whole cell protein lysates obtained from 100–150 mg of mycelium, ground in 1 mL NB-Hepes buffer (20 mM Hepes, pH 7.5, 150 mM NaCl, 50 mM K(OAc), 2 mM Mg(OAc)_2_, 1 mM dithiothreitol, 5% glycerol, and 0.1% (vol/vol) IGEPAL^®^ CA-630) and 500 µL of zirconia beads (Ø 0.5 mm, Carl Roth, Karlsruhe, Germany) in a Mini Bead Beater (5000 rpm at 4 °C, 2 × 20 s, 4 runs). The lysates were cleared via centrifugation (14.000 rpm at 4 °C for 20 min) and supernatants were analyzed by SDS-PAGE and Western blotting, performed using PAP (Sigma Aldrich cat. no. P1291) antibodies. Expression of the in locus *ct*Nup82-FpA fusion protein was verified with an α-*ct*Nup82 antibody [17] used in a 1:750 dilution in PBS/5% milk in combination with a goat-α-rabbit HRP-coupled secondary antibody (BIO RAD, Feldkirchen, Germany, cat. no. 170-6515) in a 1:3000 dilution in PBS/5% milk.

Tandem-affinity purification of fusion proteins was essentially performed as described previously [17]. In brief, the strains were cultivated in 2 l CCM medium and incubated at 50 °C with agitation at 90 rpm for 18 h. After harvesting and washing, the mycelium was dried and ground to a fine powder in a cryogenic cell mill (Retsch MM400, Haan, Germany). The material was resuspended in 40 mL NB-Hepes buffer and proteins were subsequently affinity purified from the cleared supernatants using IgG–Sepharose suspension (IgG–Sepharose 6 Fast Flow, GE Healthcare, Freiburg, Germany) and eluted by the TEV protease. In the second affinity purification step the proteins were immobilized on anti-Flag affinity gel (Sigma–Aldrich cat. no. A2220) and eluted with Flag peptides (Sigma–Aldrich cat. no. F3290). Eluates were separated by SDS-PAGE on pre-cast NuPAGE Bis Tris gradient gels (4–12% polyacrylamide) using a MOPS running buffer and analyzed by colloidal Coomassie brilliant blue G250 staining (ROTI^®^ Blue, Carl Roth).

## Figures and Tables

**Figure 1 ijms-23-03198-f001:**
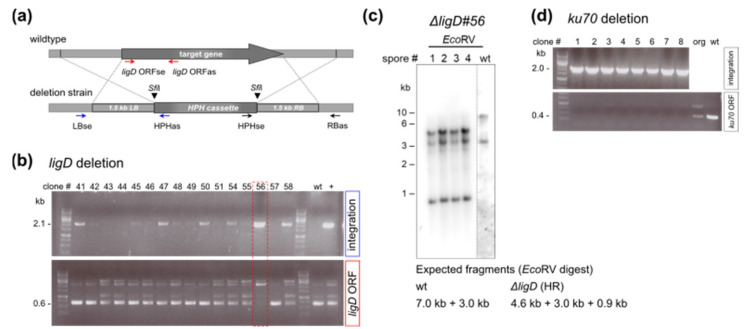
Generation of *ligD* and *ku70* deletion mutants in *C. thermophilum*. (**a**) Schematic representation of the linear PCR product used for transformation and integration of the hygromycin resistance cassette into the target gene locus via homologous regions. Arrows depict oligonucleotides used for PCR analysis. (**b**) PCR analysis performed on gDNA isolated from individual HPH-resistant Δ*ligD* transformants to verify in locus integration events. A marker-specific oligonucleotide and an oligonucleotide designed to anneal outside the homologous region flanking the HPH-cassette were used and only those clones that underwent a correct recombination event would produce an amplicon of the designated size of 2.1 kb (upper panel). Conversely, a set of oligonucleotides annealing inside the *ligD* open reading frame is not expected to result in an amplicon in case of a gene replacement event, whereas a 0.6 kb fragment is expected from the wild-type *ligD* allele (lower panel). Wild-type gDNA was included as negative control. gDNA isolated from a heterokaryotic clone from a previous transformation served as positive control for the PCR reaction. (**c**) Southern analysis confirming the deletion of *ligD*. A combination of left and right homologous flanking regions was used as probe and the expected fragments for the wild-type, and the strains undergoing homologous integration events with the HPH-resistance cassette in the *ligD* locus, respectively, are indicated below. (**d**) PCR analysis to verify *ku70* deletion was performed as described in (**b**).

**Figure 2 ijms-23-03198-f002:**
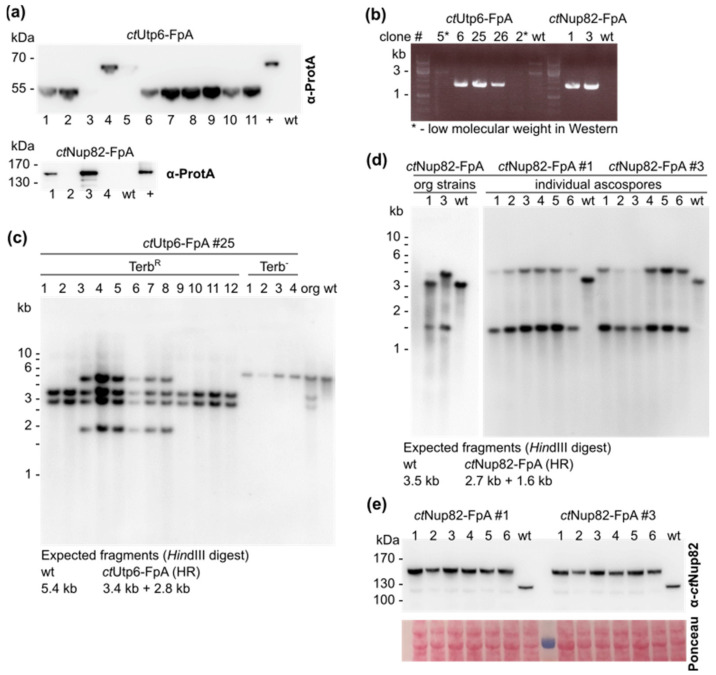
Verification of HR-mediated gene replacement events. (**a**) Representative Western analysis of individual Terb^R^ transformants showing expression of FpA-tagged Utp6- (upper panel) and Nup82-fusion proteins (lower panel) integrated in locus in Δ*ligD#56.2.* Expression was revealed with α-Protein A antibodies on whole cell protein lysates. Wild-type *C. thermophilum* protein lysate was included as negative control; the respective strain derived upon ectopic chromosomal integration of the respective fusion protein served as positive control. A molecular weight standard with molecular weights (kDa) is depicted on the left. (**b**) PCR analysis performed on gDNA isolated from Terb^R^ mycelia previously tested positive via Western analysis to verify in locus gene-replacement events. A marker-specific oligonucleotide and an oligonucleotide designed to anneal outside the homologous region were used and only those clones that underwent a correct recombination event would produce an amplicon of the designated size. For Utp6-FpA clones showing lower molecular weight bands in the Western blot, no amplicon could be produced, indicating unspecific recombination events. (**c**) Southern analysis confirming HR-mediated gene replacement in the *utp6* locus. Mycelia from 12 Terb^R^ and four Terb^−^ ascospores were harvested and subjected to gDNA isolation and digestion with restriction endonucleases. gDNA from one positively tested original strain Δ*ligD Utp6-FpA* #25 as well as wild-type gDNA was included as control. A combination of left and right homologous flanking regions was used as probe, and the expected fragments upon *Hin*dIII digestion are shown below, indicating segregation of the different alleles upon formation of sexual ascospores. (**d**) Southern analysis confirming the segregation of wild-type- and HR-mediated gene fusion alleles of Δ*ligD Nup82-FpA* #1 and #3 upon the generation of sexual ascospores. *Hin*dIII-digested gDNA of the original strains is indicated on the left, gDNA isolated from mycelia of six individual ascospores is shown on the right. Wild-type gDNA is included as control. A combination of left and right homologous flanking regions was used as probe and the expected fragments for the wild-type and the strains undergoing homologous recombination events are indicated below (the calculated fragments for the HR allele differ from the actually observed fragments, possibly due to erroneous genome annotation.) (**e**) Western analysis of mycelia resulting from individual ascospores of Δ*ligD Nup82-FpA* to verify HR-derived gene replacements leading to isogenic strains and expression of the respective fusion proteins. Expression was revealed with α-*ct*Nup82 antibodies on whole cell protein lysates and wild-type protein lysate was included as control. Equal loading was confirmed by Ponceau S staining. A molecular weight standard with molecular weights (kDa) is depicted on the left.

**Figure 3 ijms-23-03198-f003:**
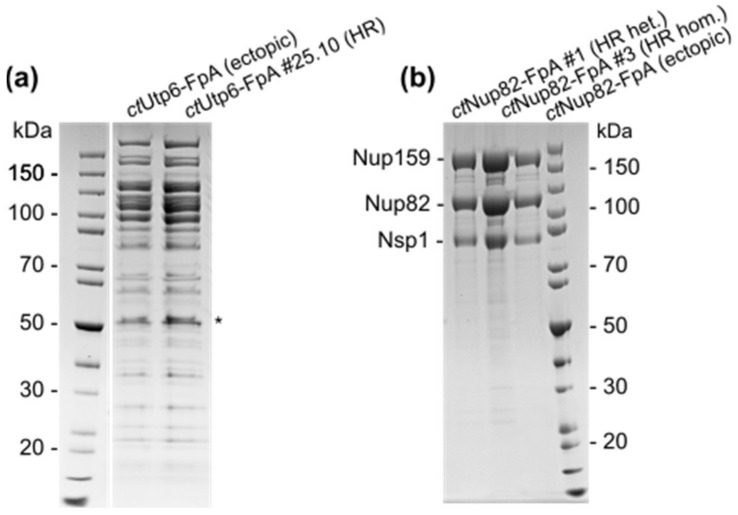
Affinity-purification of in locus C-terminally tagged fusion proteins. Tandem-affinity purification of HR-derived in locus expressed Utp6-FpA- (**a**) and Nup82-FpA fusion proteins (**b**) in comparison to their respective plasmid-based ectopically integrated fusion proteins. The 90S pre-ribosomal particle and native Nup82-subcomplex were purified, respectively. Flag-eluates were analyzed by SDS-PAGE and Coomassie staining. *—bait protein.

**Figure 4 ijms-23-03198-f004:**
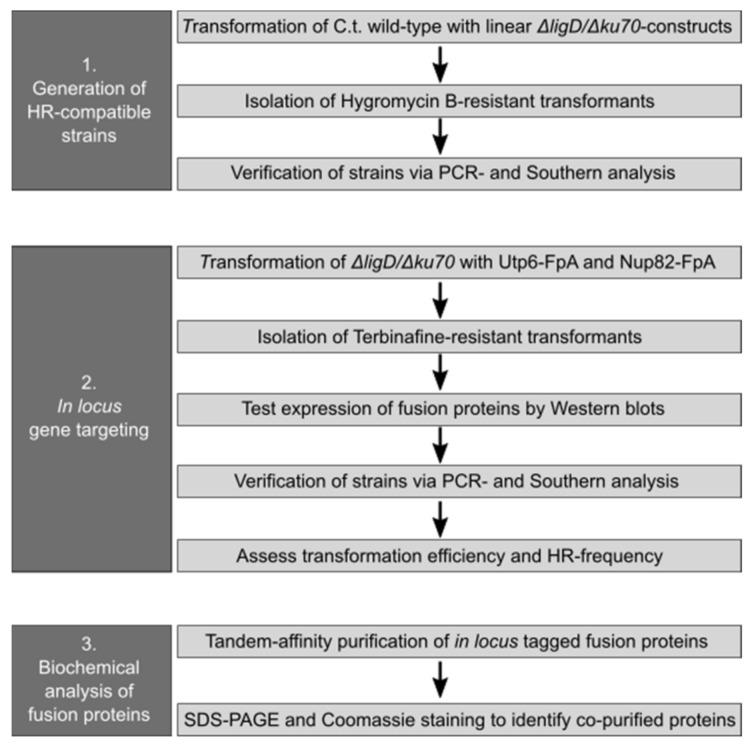
Schematic overview of the experimental steps needed to generate, verify and analyze HR-derived strains in *C. thermophilum* obtained by in locus gene targeting.

**Table 1 ijms-23-03198-t001:** Genetically modified *C. thermophilum* strains used in this study.

Strains	Genotype	Reference
Δ*ligD*	Δ*ligD::hph*	This study
Δ*ku70*	Δ*ku70::hph*	This study
Utp6-FpA (in locus)	Δ*ligD::hph, utp6:FpA, erg1*	This study
CT13	*P_UTP6_:UTP6:FpA, erg1*	[18]
Nup82-FpA (in locus)	Δ*ligD::hph, nup82:FpA, erg1*	This study
CT7	*P_NUP82_:NUP82:FpA, erg1*	[17]

**Table 2 ijms-23-03198-t002:** Oligonucleotides used in this study.

Oligonucleotides	Sequence (5′-3′ Direction)	Purpose
ligD_lb_se	CAACTTCATGCCGAACGTTCTCGAG	*ligD* deletion
ligD_lb_asSfiI	ATAggccatctaggccGGTTGATGGGGTCGAACTCGGG	*ligD* deletion
ligD_rb_seSfiI	GTGggcctgagtggccTGATTTTATGGAGCGCTAAGGGAGG	*ligD* deletion
ligD_rb_as	CCTGCAGCTTGCTGATGAGCTTC	*ligD* deletion
ligD_nested_se	CGGTCGGAAGACCGTAACCAGC	*ligD* deletion
ligD_nested_as	CGTTCTGTGACCTGACCTCTTGTC	*ligD* deletion
ligD_ORFse	ATGAGCGCCACAAAGAAGAGGAC	Analytical PCR
ligD_ORFas	GAAATCGCCGGCCGTACCAGC	Analytical PCR
HPHcheck_as	CGCGGTGAGTTCAGGCTTTTTCAT	Analytical PCR
HPHcheck_se	GCACTCGTCCGAGGGCAAAGG	Analytical PCR
ku70lb_se	CTCGGCTTGGTATACATCGGCCG	*ku70* deletion
ku70lb_asSfiI	ATAggccatctaggccGGTTGCTGGTAAGTGTGATGTCGTTC	*ku70* deletion
ku70rb_seSfiI	GTGggcctgagtggccTGATTCGTCATGATTATGTGTAATGCCTTG	*ku70* deletion
ku70rb_as	GGATTTGCCGTCCAGTGGGACC	*ku70* deletion
ku70nested_se	GCCAAACGACTTGCCTGTAGGG	*ku70* deletion
ku70nested_as	GTTCTGGGCTTCGTCTCCTTTGG	*ku70* deletion
ku70_ORF_se	ATGGCGTACGGTGATGACGATG	Analytical PCR
ku70_ORF_as	CAGGTCAACCAACAGGTAGCAG	Analytical PCR
ku70_LB_out	CAACAGTCGCCCTCAATTGCTC	Analytical PCR
utp6fus_lb_se	GACGAAATTCGCAACCTCGTAGCC	Utp6 fusion
utp6fus_lb_asSfiI	ATAggccgcgttggcccgGCTGGGCGACTCCTCCAGCG	Utp6 fusion
utp6_rb_seSfiI	GTGggcctgagtggccTCGTTTGGCTGCAAAGAGGGTCAAAATC	Utp6 fusion
utp6_rb_as	CCTAAGATCCCTCCCGAGGTATTAG	Utp6 fusion
utp6fus_nest_se	GGCACCAAGCCGACCGATTTCC	Utp6 fusion
utp6fus_nest_as	CAGTGCCCACGACTATGACCACAC	Utp6 fusion
nup82fus_lb_se	CGAAGATGAACAGGACGATGAGGTG	Nup82 fusion
nup82fus_lb_asSfi	ATAggccgcgttggcccgCCCAATGCTGAGCCTCTCGATTC	Nup82 fusion
nup82fus_rb_seSfi	GTGggcctgagtggccGTATGGGAACCCGAGAATATGGAAG	Nup82 fusion
nup82fus_rb_as	GAAGTCTGCATCTAGGTGGTTCTG	Nup82 fusion
nup82fus_nest_se	GCCAAGGGTTGCGGACCTGATG	Nup82 fusion
nup82fus_nest_as	CTTTAATCTGAGCGCGCAGATCAAC	Nup82 fusion

## Data Availability

Not applicable.

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
