# Peer review of "A Homologous Recombination System to Generate Epitope-Tagged Target Genes in Chaetomium thermophilum: A Genetic Approach to Investigate Native Thermostable Proteins"

_ijms, 2022, doi:10.3390/ijms23063198_

Round 1

Reviewer 1 Report

The authors aimed to develop a homologous recombination (HR)-compatible C. thermophilum strain to be able to conceive precise gene modification/replacement experiments, as HR allows efficient incorporation, expression control, and modification of desired target genes. The manuscript is well written, scientific significant and can be accepted for publication.

Minor changes and suggestions:

  1. Please provide the information of the proteins "69% identity and 79% similarity with the C. globosum LigD protein. To obtain a ligD deletion mutant in C. thermophilum (Lines 82-83).
  2. Figures 2C and 2D can be cropped to use less space? 
  3. "Next, we wanted to analyze the respective affinity-purified protein complexes..." should read "The respective affinity-purified protein complexes  were analyzed' (Line 250).
  4. The authors can provide more information "Eluates were  separated by SDS-PAGE and analyzed by Coomassie staining." Line 431).

Reviewer 2 Report

The paper focusing on Chaetomium thermophilum wants to describe the development of a homologous recombination system to exploit the derived thermostable fusion proteins for tandem-affinity purification. The paper is well written and organized even if frequently the authors consider their work only for specialists. Instead is of high importance involving in reading also less skilled operators by giving more particulars and perspectives in using this approach.

In this perspective, I suggest the authors change vague terms in the Introduction as “various” into a few examples in order to increase the interest in this species and biotech exploitation (please see below). In the meanwhile, possible applications can be used as keywords for the readers’ craving…even if the paper is not focused on a specific biotech application, could be an interesting preparatory step. Chaethomium thermophilum being part of the title is already indexed and should be substituted.

Even the title could be more explicative changing it in something like “ Investigation on thermostable proteins: a homologous recombination system to generate….”

A workflow /synopsis of the steps listed in the material and methods section is basic for study design understanding and should be added.

Below are some additional minor notes

The citation in the text should follow the journal’s guidelines, and sentences shouldn’t be trunked (e.g., described in [43].)

L377-378 the sentence should be shortened like that “Wild-type Chaetomium thermophilum var. thermophilum (La Touche) DSMZ 1495 was used [42].

You don’t need to explain what DSMZ means because is a worldwide recognized culture collection as ATCC and CBS, and the following number is the strain identifier nr. More, the reference should be placed just before punctuation…but this one can be removed because is the collection that grant for identification

L381 Concisely… please add more particulars is too concise for an external reader

L386 hygromycin B and terbinafine suppliers

Table 1 the title should be modified because they are “transformed” strains. the column header should be in the plural form: strains

L390 described in [43]. Please modify in a more readable way, e.g., …described by Sambrook and colleagues [43]. The same at L392 etc. Please check this item along with the manus.

L52  to be able to conceive  ...that sounds wordy. Please simplify
